# Quality of Life and Emotional Problems of COVID-19 Patients after Discharge: A One-Month Longitudinal Study

**DOI:** 10.3390/healthcare12040488

**Published:** 2024-02-17

**Authors:** Chiu-Feng Wu, Ting-Yun Lin, Sheng-Kang Chiu, Chu-Hsuan Cheng, Wen-Lin Su

**Affiliations:** 1Department of Nursing, Taipei Tzu Chi Hospital, Buddhist Tzu Chi Medical Foundation, New Taipei City 23142, Taiwan; chiu-feng@tzuchi.com.tw (C.-F.W.); forcechoas@gmail.com (C.-H.C.); 2Division of Nephrology, Taipei Tzu Chi Hospital, Buddhist Tzu Chi Medical Foundation, New Taipei City 23142, Taiwan; tingyunlin@tzuchi.com.tw; 3Division of Infectious Diseases, Department of Internal Medicine, Taipei Tzu Chi Hospital, Buddhist Tzu Chi Medical Foundation, New Taipei City 23142, Taiwan; csk33kimo@hotmail.com; 4Division of Pulmonary and Critical Care Medicine, Department of Internal Medicine, Taipei Tzu Chi Hospital, Buddhist Tzu Chi Medical Foundation, New Taipei City 23142, Taiwan

**Keywords:** COVID-19, quality of life, mental health, depression, anxiety, stress

## Abstract

Aim: The first coronavirus disease 2019 (COVID-19) outbreak in Taiwan occurred in May 2021 and many individuals were infected. All COVID-19 patients were quarantined in designated facilities until they fully recovered to prevent the spread of the disease. Prolonged quarantine could adversely affect these patients. In this study, we focused on investigating changes in the quality of life and mental health of individuals discharged from hospital after recovering from COVID-19. Methods: This study employed a longitudinal design and surveyed individuals discharged from a teaching hospital in northern Taiwan in 2021 within one week of their discharge and again after one month. An online questionnaire comprising the participants’ background, respiratory function (COPD Assessment Test), quality of life (WHOQoL-BREF), and emotional problems (DASS-21) was administered to the participants. Results: A total of 56 participants actively took part in both surveys. We observed that participants with abnormal respiratory function had a lower physical and psychological quality of life, especially those with severe symptoms requiring endotracheal intubation during the treatment period of COVID-19. Additionally, approximately 30% of participants experienced anxiety problems throughout this study period. Finally, patients with COVID-19 symptoms exhibited a lower quality of life and higher levels of severe emotional problems. Conclusions: According to our findings, it is necessary to monitor and provide appropriate interventions for individuals who have recovered from COVID-19, especially those who experienced severe symptoms that required endotracheal intubation during COVID-19 treatment. These interventions, such as symptom management and psychological support, can help improve their quality of life and reduce emotional problems. Therefore, after the participants are discharged, hospitals should regularly track the patients’ status and provide appropriate support or referrals to help these individuals. Otherwise, future research could include more participants and follow up with them for longer to investigate the longitudinal impact of COVID-19.

## 1. Introduction

Coronavirus disease 2019 (COVID-19) is caused by severe acute respiratory syndrome coronavirus 2 (SARS-CoV-2); the symptoms are similar to the flu. However, the difference is that more individuals infected with COVID-19 experience chest pain, chest tightness, difficulty breathing, and a loss of taste or smell compared to those infected with the flu. Additionally, the symptoms of COVID-19 can continue for an extended period [1].

Most symptoms of COVID-19 infection were either not apparent or of mild severity [2], and the virus is transmitted through air particles. This made it easy to spread, and the disease quickly became a pandemic [3]. Globally, according to a report by the World Health Organization (WHO), up until the end of June 2021 COVID-19 had infected 180 million individuals, and caused about 4 million people’s deaths [4].

In Taiwan, the COVID-19 local outbreak began in May 2021 and continued until 30 June 2021, during which time 14,748 individuals were infected [5]. Due to inadequate medical resources to combat COVID-19, the Taiwanese government quarantined all asymptomatic individuals and those with mild symptoms, while quarantining patients with severe symptoms in dedicated hospital wards. Quarantined patients with COVID-19 were only released after successfully passing highly sensitive COVID-19 screening or they remained under continuous quarantine in a quarantine institution until they met the screening criteria [6]. These measures were aimed at preventing the spread of the disease.

According to the definition of health by the WHO in 1946, health is a state of complete physical, mental, and social well-being, and not merely the absence of disease or infirmity; health-related quality of life is an indicator that reflects a person’s health condition [7]. However, during COVID-19, inpatients’ three health dimensions could be adversely affected by the disease symptoms and quarantine measures they experienced. Some studies have reported that COVID-19 inpatients experience fear and shock due to their own disease [8,9], a mental burden resulting from their quarantine by others [8,9], and concerns about transmitting the disease to others with whom they had contact before quarantine [9,10]. These problems affect these patients’ mental health and quality of life [11,12].

However, the status of the patients who recovered from COVID-19 after their release from quarantine remains unclear. Several studies have reported the adverse effects of governmental anti-epidemic measures for COVID-19 on the general public [13,14]. In some countries, people were required to isolate in their homes, leading to people having some physical or mental health problems [13,14]. In Taiwan, while most individuals were not mandated to quarantine at home, there were strict restrictions on outdoor activities for epidemic prevention, resulting in the closure of some public places, impacting their mental health [15]. Discrimination towards and fear of individuals at a high risk of COVID-19 infection was also observed due to a lack of understanding of the disease [16]. A study also reported that people at high risk of infection experienced discrimination, which impacted their mental health [17]. Notably, some individuals recovering from COVID-19 may experience persistent symptoms [1]. These experiences can affect the mental health and quality of life of individuals who have recovered from COVID-19.

Therefore, this study aimed to investigate the changes in the quality of life and mental health of individuals who have recovered from COVID-19. The findings of this study can serve as a valuable reference for combating emerging diseases and enhancing the quality of care and well-being of patients.

## 2. Methods and Materials

The target population consisted of individuals who had been out of quarantine for less than a week and had been quarantined in the quarantine hotel or dedicated ward of a teaching hospital located in New Taipei City, in 2021. In terms of exclusion criteria, this study required participants to answer the questionnaire independently. Therefore, individuals who could not read or respond to the online questionnaire independently were excluded. Additionally, due to legal restrictions, individuals under 20 years of age were not permitted to answer the questionnaire independently and were also excluded.

This study utilized a longitudinal design, collecting participants’ data through an online questionnaire shortly after they came out of quarantine, after less than a week, and another a month after completing the first questionnaire. In addition, to enhance participants’ willingness to fully participate in the study, we provided a gift certificate worth TWD 100 as payment, which served as compensation for transportation costs when participants had to move to an area with Internet access to answer the online questionnaire. TWD 100 (about USD 3.2) is a small amount in Taiwan. An individual taking a taxicab to an area with Internet access would need at least TWD 190 for a round trip. Additionally, according to Stanley et al. [18], an incentive value of up to USD 5 can significantly increase the response rate to a survey with minimal impact on data quality or bias. Therefore, we believe that our incentive to participants would not introduce bias to our survey data.

### 2.1. Research Tools

Participant data were collected through an online self-administered questionnaire consisting of four parts: participant background, respiratory function, quality of life, and emotional problems.

1. Participant background: This part collected participants’ demographic variables and their quarantine or hospitalization conditions, including age, sex, whether they were hospitalized during their quarantine period, current presenting COVID-19 symptoms, chronic diseases, duration of quarantine, use of oxygen supply cartridges during hospitalization, tracheal intubation during hospitalization, and treatment in the intensive care unit (ICU).

2. Respiratory function: The participant’s respiratory function was measured using the Chronic Obstructive Pulmonary Disease (COPD) Assessment Test (CAT), developed by Jones et al. [19]. This self-administered instrument was initially designed to assess the severity of COPD symptoms, but it can also be used to evaluate the recovery status of patients with COVID-19 [20]. Additionally, this instrument is a brief and commonly used scale; therefore, we chose it as our research instrument. Regarding reliability, a review article from 2014 reported that Cronbach’s α was 0.85 and 0.98 [21], and a current study reported it as 0.924 in 2023 [22], indicating good internal consistency. The test–retest reliability ranged from 0.80 to 0.96, demonstrating stability over time. The instrument consisted of eight items, each featuring two paragraphs at either end. The participants selected the score that best matched their own situation, ranging from 0 to 5. The total score ranges from 0 to 40, with a higher score indicating a more significant impact on their quality of life due to COPD symptoms. Specifically, scores between 0 and 10 represent a mild impact, score between 11 and 20 denote a moderate impact, scores between 21 and 30 indicate a moderately severe impact, and scores between 31 and 40 represent a severe impact [19]. In this study, the official version of traditional Chinese was used, and a score of 10 or higher was employed as the criterion for identifying abnormal respiratory function, as recommended in Daynes et al. [20].

3. Quality of life: The participants’ quality of life was measured using The World Health Organization Quality of Life (WHOQoL-BREF), developed by the WHO and translated into its traditional Chinese version for Taiwan by Yao et al. [23]. The questionnaire’s content was adjusted to align with Taiwan’s local culture, including social and environmental aspects. This allows for a more comprehensive assessment of the quality of life of discharged individuals than other instruments; therefore, we chose it as our research instrument. The traditional Chinese version for Taiwan of the WHOQoL-BREF comprises 28 items (26 original WHOQoL items and two items specific to Taiwan’s local culture). It can be separated into six domains (overall quality of life, general health perceptions, physical health, psychological health, social relationships, and environment). Participants used a five-point Likert scale to rate each item, with scores ranging from 1 (not at all) to 5 (extremely), based on the similarity between the item description and their situation. Each domain was converted to a 0–100 score. In terms of reliability, the Cronbach’s α of the general instrument was 0.91, and each domain was between 0.70 and 0.77 in the original study [23], showing good reliability.

4. Emotional problems: The participants’ emotional problems were measured using the Depression Anxiety Stress Scales-21 (DASS-21). This instrument is a brief version developed by Antony et al. [24] based on Clark and Watson’s [25] theoretical model, which reformulated the original 42-item scale by Lovibond [26]. The DASS-21 comprises 21 items in three domains: depression, anxiety, and stress. Participants rated each item on a scale from 0 (did not apply to me at all) to 3 (applied to me very much or most of the time) based on the similarity between the item description and their own situation over the past week. Regarding the severity levels, depression is classified into five levels: normal (0–9), mild (10–13), moderate (14–20), severe (21–27), and extremely severe (28+). Anxiety is also categorized into five levels: normal (0–7), mild (8–9), moderate (10–14), severe (15–19), and extremely severe (20+). Similarly, stress levels are divided into five categories: normal (0–14), mild (15–18), moderate (19–25), severe (26–33), and extremely severe (34+). For the purposes of this study, we considered scores falling within the mild or severe range in each domain of the scale as indicative of abnormal emotional problems. Regarding reliability, the Cronbach’s α coefficient ranged from 0.87 to 0.94 in the original study, indicating good internal consistency [24]. This scale has an official traditional Chinese version [27], and is relatively easy to obtain; therefore, it was used in our study.

### 2.2. Data Analysis

The following analytical methods were used in this study.

First, the differences between participants who fully participated in this study and those who only participated in the first survey were analyzed using independent *t*-tests or chi-square tests, depending on whether the data were continuous or categorical.

Second, changes in the participants’ variables and outcomes between the first and second surveys were analyzed using paired *t*-tests or chi-square tests, depending on whether the variables were continuous or categorical.

Third, the relationship between a participant’s background, quality of life, and emotional problems was analyzed using correlation coefficients if the participant’s background variables were continuous, and analyzed using independent *t*-tests or one-way analysis of variance (ANOVA). 

Furthermore, since this research was longitudinally designed, the Generalized Estimating Equations (GEE) method is recommended. GEE is similar to regression analysis but can adjust for matched data, such as different time points of follow-up. It is used to provide a predictive model with matched data, adjust for confounding variables, and identify the variables that affected the participants’ quality of life or emotional problems. In this study, we adjusted for the follow-up wave, participant’s age, and sex. Additionally, we included all variables that significantly affected the outcomes (WHOQoL or DASS-21), as identified by independent *t*-tests, one-way ANOVA, and correlation coefficients.

Finally, the analysis software used in this study was R 4.1.3 (R Foundation for Statistical Computing, Vienna, Austria) [28].

### 2.3. Ethical Considerations

The Institutional Review Board of the teaching hospital approved this study (approval no. IRB:10-XD-090, approved on 22 July 2021). Regarding the licensing of these instruments, we obtained permission from the respective authors of the Taiwan traditional Chinese versions of the WHOQoL-BREF and CAT. Furthermore, DASS-21 is a freely available resource [29]. During the participant recruitment phase, participants were fully informed of the purpose and content of the study by trained researchers to ensure that the information obtained by the participants was consistent, and that they understood how to answer the online questionnaire. Informed consent messages were also presented on the first page of the online questionnaire, and the participants were required to read all messages and agree to participate before answering the questionnaire. This survey was anonymous, and individuals and participants were allowed to refuse or leave this study without needing to provide a reason. Refusing or leaving this study also did not impact their rights and interest in medical care. The participants’ personal data (email address and telephone number) were encrypted using the R language package, allowing for data matching while ensuring security.

## 3. Results

A total of 96 participants completed the questionnaire. Only 56 participants completed the final questionnaire during the follow-up survey. The flow chart in Figure 1 shows this

The differences between participants who completed all the surveys and those who only participated in the first survey are shown in Table 1. We found that the two groups were not significantly different regarding their demographic variables. However, participants who only participated in the first survey had a slightly significantly higher score in their general health perceptions (participants in the whole study: 3.35 ± 0.84, participants in the first survey only: 3.70 ± 0.82, *p* < 0.05) and a slightly significantly lower score in their level of stress (participants in the whole study: 8.03 ± 6.99, participants in the first survey only: 5.20 ± 5.23, *p* < 0.05). Therefore, the participants who withdrew could be individuals who are relatively healthy and experienced fewer mental health problems. This could potentially lead to a slight overestimation of the severity of the problems related to participants’ quality of life or mental health in our results.

### 3.1. Changes between the First and the Second Surveys

The changes between the first and second surveys are presented in Table 2. In terms of the demographic variables, participants’ acute symptoms significantly decreased in the second survey (first survey: had one symptom = 35.71%; had at least two symptoms = 26.78%. In the second survey: had one symptom = 21.42%, had at least 2 symptoms = 14.28%, *p* < 0.05); however, their respiratory function did not significantly increase. This indicates that while most participants’ COVID-19 symptoms improved after the second survey, their respiratory function problems persisted.

Concerning the scales, participants’ overall quality of life, as measured by the WHOQoL, significantly increased in the second survey (first: 3.54 ± 0.69, second: 3.80 ± 0.64, *p* < 0.05), indicating that participants perceived an improvement in their quality of life in the second survey. In other domains of quality of life, the participants reported feeling improvement, although the difference was not statistically significant. Furthermore, we observed a higher proportion of participants experiencing anxiety problems, and, even after one month, the proportion did not show a significant decrease.

### 3.2. Results of the Relationship between the Background Variables of the Participants and their Scale Scores

In the results of the correlation coefficients, participants’ age showed no significant effect on any of the scales (*p* > 0.05), indicating that the participants’ age did not affect their quality of life and mental health. However, between quality of life and mental health, both scales showed a significant (*p* < 0.001) mild to strong correlation (r = −0.26 ~ −0.64), suggesting that a better quality of life is associated with fewer mental health problems.

In the relationship between the variables and scale scores, sex and being hospitalized during the quarantine duration did not affect the scores of scales (*p* > 0.05).

In terms of the numbers of chronic diseases, the participant with more than two chronic diseases (2.50 ± 0.54) had general health perceptions scores lower than those of the participants with less than two chronic diseases (0: 3.49 ± 0.82, 1: 3.56 ± 0.73, *p* < 0.05). In addition, participants with more than two chronic diseases (54.02 ± 26.00) had physical quality of life scores that were lower than those with no chronic diseases (74.45 ± 12.17, *p* < 0.05). Therefore, participants with more symptoms of chronic diseases felt that they were in poorer health, which also appeared in the physical domain.

In the number of current COVID-19 symptoms, the participants with more than two COVID-19 symptoms (2.93 ± 0.96) had general health perceptions scores lower than the participants with no symptoms (3.71 ± 0.78, *p* < 0.05), and also lower scores in their physical of quality of life (0: 76.36 ± 11.04, 2: 60.48 ± 21.95, *p* < 0.05). In terms of mental health, participants with more than two COVID-19 symptoms had more severe problems with anxiety (0: 3.05 ± 4.13, 2: 9.47 ± 9.58, *p* < 0.05) and stress (0: 5.81 ± 5.72, 2: 11.60 ± 8.63, *p* < 0.05) than those with no symptoms. Therefore, participants with more COVID-19 symptoms felt they were in poorer health than those with no symptoms. Additionally, they also experienced more anxiety and stress than those with no symptoms.

In terms of receiving treatment in the ICU during their hospitalization period, participants who received treatment in the ICU had lower general health perceptions than those who did not (no: 3.45 ± 0.84, yes: 2.71 ± 0.49, *p* < 0.05), indicating that participants receiving treatment in the ICU during hospitalization felt they were in poorer health than those who did not.

In the use of oxygen supply cartridges during hospitalization, participants who received oxygen supply cartridges had a lower overall quality of life (no: 3.58 ± 0.70, yes: 3.39 ± 0.65, *p* < 0.05), general health perceptions (no: 3.56 ± 0.73, yes: 2.69 ± 0.86, *p* < 0.05), and physical quality of life (no: 74.67 ± 12.12, yes: 57.97 ± 22.22, *p* < 0.05) than those who did not. Therefore, participants who had received oxygen supply cartridges during their hospitalization period felt they were in poorer health than those who did not receive oxygen.

Participants who had received tracheal intubation treatment during hospitalization had a lower physical (no: 72.44 ± 14.36, yes: 41.67 ± 27.04, *p* < 0.05) and psychological (no: 65.33 ± 14.24, yes: 47.22 ± 2.41, *p* < 0.001) quality of life than those who did not receive tracheal intubation. Therefore, participants who received tracheal intubation treatment had lower physical and mental health than those who did not receive it.

Finally, the participants with abnormal respiratory function had a lower quality of life (overall quality of life: normal = 3.69 ± 0.72, abnormal = 3.07 ± 0.27, *p* < 0.001; general health perceptions: normal = 3.62 ± 0.66, abnormal = 2.57 ± 0.85, *p* < 0.001; physical health: normal = 76.19 ± 9.91, abnormal = 54.59 ± 21.40, *p* < 0.001; psychological health: normal = 68.35 ± 12.98, abnormal = 52.38 ± 12.09, *p* < 0.001; social relationships: normal = 66.82 ± 13.63, abnormal = 56.25 ± 13.43, *p* < 0.05; environment: normal = 68.12 ± 13.19, abnormal = 59.33 ± 15.70, *p* < 0.05) and experienced more anxiety (normal = 4.57 ± 5.06, abnormal = 10.57 ± 9.53, *p* < 0.05) and stress (normal = 6.81 ± 5.66, abnormal = 11.71 ± 9.31, *p* < 0.05) problems.

### 3.3. Results of Analysis by Generalized Estimating Equations

Participants’ age, sex, and the variables that significantly affected their scores on the scales were included in the GEE model to analyze which variables could predict their quality of life and emotional problems (Table 3 and Table 4). However, the depression domain of the DASS-21 was not analyzed because none of the variables significantly affected the depression scores. The results showed that age, the number of chronic diseases, current COVID-19 symptoms, tracheal intubation during hospitalization, and respiratory function abnormalities predicted the quality of life and emotional problem scores.

Regarding participants’ ages, the older the participants, the better their general health perceptions scores (β = 0.012 ± 0.004, *p* < 0.05), indicating that older participants feel their general health is better than that of the younger participants. Concerning the number of chronic diseases, participants with more than two chronic diseases had lower scores in their general health perceptions (β = −0.51 ± 0.20, *p* < 0.05) and physical health (β = −10.97 ± 4.73, *p* < 0.05) in the WHOQoL compared to participants with no chronic diseases. Therefore, participants with more than two chronic diseases perceived their general health and physical health as worse than those with no chronic diseases.

Regarding the number of acute COVID-19 symptoms, participants who still had acute COVID-19 symptoms exhibited lower general health perceptions than those with no acute COVID-19 symptoms (one symptom: β = −0.34 ± 0.13, *p* < 0.05; two or more symptoms: β = −0.63 ± 0.14, *p* < 0.001). Additionally, participants with more than one acute COVID-19 symptom had lower physical health (β = −13.77 ± 3.27, *p* < 0.001) and higher anxiety scores (β = 3.65 ± 1.34, *p* < 0.05) compared to those with no symptoms. Therefore, the more acute COVID-19 symptoms participants have, the worse they perceive their general health and physical health, and the more severe their anxiety problems.

Regarding participants who underwent tracheal intubation during hospitalization, those who received tracheal intubation exhibited lower physical health (β = −21.26 ± 4.76, *p* < 0.001) and psychological health (β = −17.56 ± 4.75, *p* < 0.001) in terms of their quality of life compared to those who did not. Therefore, participants who underwent tracheal intubation during hospitalization for COVID-19 had lower physical and psychological health than those who did not, regardless of whether it was in the first or second wave of the survey.

Regarding respiratory function, participants whose CAT scores were abnormal had lower scores for their quality of life (overall quality of life: β = −0.64 ± 0.11, *p* < 0.001; general health perceptions: β = −0.66 ± 0.15, *p* < 0.001; physical health: β = −15.48 ± 3.48, *p* < 0.001; psychological health: β = −17.07 ± 2.75, *p* < 0.001; social relationships: β = −10.22 ± 3.09, *p* < 0.001; environment: β = −11.45 ± 3.55, *p* < 0.001) and higher scores for anxiety (β = 5.88 ± 1.78, *p* < 0.001) and stress (β = 5.98 ± 2.11, *p* < 0.05) compared to participants with normal CAT scores. Therefore, participants with abnormal respiratory function had a lower quality of life and experienced more severe anxiety and stress problems.

### 3.4. Summary of Results

In summary, after being discharged for about one month, participants’ overall quality of life significantly improved. However, in other domains and in terms of participants’ mental health problems, although the scores increased, the changes were not statistically significant. In addition, we found that about 30% of participants in the first survey had anxiety problems, but in the second survey the percentage did not significantly decrease.

In the results of analyzing the relationship between the variables and scale scores, we found that participants with more chronic diseases, those who had received treatment in the ICU during hospitalization, those who had received oxygen supply cartridges, those who had undergone tracheal intubation during hospitalization, those with more acute COVID-19 symptoms, and those with abnormal respiratory function had a lower quality of life and more problems with anxiety and stress. After adjusting for the GEE model, participants with more chronic diseases, those who had received oxygen supply cartridges, those who had undergone tracheal intubation during hospitalization, those with more acute COVID-19 symptoms, and those with abnormal respiratory function still had a lower quality of life or experienced more severe anxiety or stress problems.

## 4. Discussion

The objective of this study was to investigate the changes in the quality of life and mental health among individuals discharged from the hospital after recovering from COVID-19.

An interesting finding is that a high percentage of participants experienced anxiety problems even after being discharged for one month. Additionally, participants with severe symptoms (requiring treatment with endotracheal intubation) during hospitalization had lower a physical and psychological quality of life. Moreover, participants who still had acute COVID-19 symptoms or had not recovered their respiratory function experienced a lower quality of life and more severe anxiety and stress problems. Interestingly, after one month, most participants with abnormal respiratory function had not recovered. Finally, participants with more chronic diseases felt their health was poor.

As there were no adequate methods for treating COVID-19 in 2021, patients with COVID-19 were quarantined until they could no longer spread the disease. During quarantine, patients could not have contact with other people except medical staff [6]. Some studies have shown that long-term quarantine has adverse effects on quality of life and psychological health [9,12,30]. These adverse effects arise from the quarantine of other people, COVID-19 symptoms, and uncertainty about the disease, ultimately affecting patients’ quality of life [8,31,32] and psychological health [9]. Deng et al. [8] surveyed COVID-19 inpatients in a hospital by telephone and reported that the psychological health problems observed were mainly related to the quarantine experienced by quarantined patients with COVID-19.

In our study, although all the scale scores showed improvement, most of these changes were not statistically significant. Additionally, in terms of quality of life, when comparing our results with the data reported in the traditional Chinese version of the Taiwan user manual [23], and considering both surveys, the scores were not lower than those of the general adult population in Taiwan. A possible reason is that our study had a small sample size, which is one of its limitations. However, some other studies could also explain our results [9,33]. One study interviewed COVID-19 patients who were quarantined in a hospital. Initially, they experienced psychological health problems, but after adapting to quarantine their psychological health improved [9]. Another study, using an online questionnaire survey, reported that social support from family and friends could alleviate the helplessness of residents affected by the prevention measures for COVID-19 [33]. In our study, both surveys were conducted after the participants were discharged, allowing them to receive direct social support from their families and friends, resulting in their quality-of-life recovery. Therefore, this could cause our participants to have a high quality of life, and thus changes between both surveys were not significant.

However, some studies have reported contradictory results [34,35,36,37]. These studies conducted a cross-sectional survey on discharged patients who had severe COVID-19 symptoms and required hospitalization in the intensive care unit. They found that some patients continued to experience problems with self-care, pain, and depression or anxiety, which affected their quality of life, even after being discharged for more than a month. In our study, most participants had a mild or moderate severity of COVID-19 symptoms, which may account for the better quality of life we observed compared to those studies. However, the results of our study, which were adjusted by the GEE model, revealed that participants with severe symptoms during hospitalization, requiring endotracheal intubation, had a lower physical and psychological quality of life. This pattern persisted in both the first and second waves of the survey, aligning with the findings of these studies.

Interestingly, more than 30% of the participants had anxiety problems in the first survey. However, in the second survey, the percentage of participants who experienced anxiety did not decrease significantly. Some studies have yielded similar results [30,34,35,36]. Hoque et al. [35] reported that symptoms of COVID-19 could affect the quality of life and emotional well-being of patients who have recovered from COVID-19. Similarly, van der Sar–van der Brugge et al. [38] reported that respiratory function could affect the quality of life and emotional well-being of individuals who have recovered from COVID-19. Our results are consistent with these findings. We found that participants with two or more COVID-19 symptoms and abnormal respiratory function experienced more severe anxiety than participants with no COVID-19 symptoms and normal respiratory function.

Additionally, similar to the report by Mizrahi et al. [1], who stated that individuals recovering from COVID-19 continue to experience persistent symptoms, our study found that, in the second survey, approximately 35% of participants still had more than one symptom of COVID-19, and 16% of participants had abnormal respiratory function. These findings may explain the high percentage of participants with anxiety problems in both surveys. In relation to this, a biopsychosocial model [39], which provides insights into the relationships among individuals’ physical, psychological, and social aspects, supports our results. It explains how participants with worse physical conditions also experience an impact on their mental health. Therefore, our participants with more symptoms of COVID-19, abnormal respiratory function, or more chronic diseases had a lower psychological quality of life or more severe mental problems. In terms of social aspects, there were some reasons that our participants’ mental health was impacted. Some studies reported that epidemic prevention measures for COVID-19 in particular districts, or those implemented by governments, could adversely affect residents’ mental health [13,40]. Other studies have found that patients who have recovered from COVID-19 or are at high risk of infection experienced discrimination during the pandemic, which impacted their mental well-being [17,41]. Caroppo et al. [13] reported that while the government’s adoption of social distancing policies effectively suppressed the spread of COVID-19, it also caused many inconveniences in the lives of residents and increased their psychological distress. Additionally, Liu et al. [42] reported that ignorance and fear of COVID-19 led to discrimination against individuals infected with the virus and even those at high risk of infection. In our study, even though the participants had been discharged, pandemic prevention measures still affected their living areas. Being infected with COVID-19 could lead to discrimination from friends or neighbors, perpetuating their anxiety problems.

In the background variables of our study, we found that participants with more chronic diseases had a lower quality of life, similar to many studies [12,43,44,45,46]. These studies suggest that patients with chronic diseases have a lower quality of life due to the limitations imposed by their chronic conditions on physical functions. O’Dwyer et al. [12] reported that COVID-19 patients with diabetes also found an impact on their social functioning, role limitations, and increased bodily pain domains of quality of life; therefore, the characteristics of different chronic diseases may have additional differential impacts on quality of life. However, since our study included a small number of participants, we cannot analyze the differences between separate chronic diseases.

In terms of participants’ ages, after adjusting for confounding factors we found that participants of older ages felt themself healthier. However, these results differ from other studies; those studies show that participants with older ages had a lower quality of life [46,47]. Nevertheless, in the results of Chen et al. [47], who surveyed COVID-19 patients in a hospital’s dedicated COVID-19 wards using a short form, it was reported that older-age COVID-19 patients had better vitality. This finding might be correlated with our results. In addition, it could also be due to the limitations of this study, as our study had a small sample.

According to our findings, it is necessary to monitor and provide appropriate interventions for individuals who have recovered from COVID-19, especially those who experience severe symptoms that require endotracheal intubation during COVID-19 treatment. These interventions, such as symptom management and psychological support, can help improve their quality of life and reduce emotional problems. A guideline (leaflets, manuals, or web page) or a caring telephone call could be effective methods for these discharged patients. A review article suggests that providing discharged patients with guidelines, including information about the patient’s medical condition, how to mitigate post-COVID-19 syndromes or prevent complications, and how to access support if needed, can effectively make patients feel comfortable [48]. In addition, Bernocchi et al. [49] propose telecare nursing through telephone calls, which has been shown to effectively monitor the health situations of these discharged patients and improve their quality of life.

### 4.1. Limitations

Our study had some limitations. First, the participants came from the same teaching hospital located in northern Taiwan; therefore, the results may not be extrapolated to other groups. Second, due to the diminishing of the first COVID-19 outbreak after August 2021, there were almost no new patients with COVID-19 in the receiving hospital until the outbreak recurred in 2022, resulting in fewer participants participating in the entire process. Third, although the demographic variables of the participants who only participated in the first survey showed no differences from those who completed the full study, their general health perceptions of their quality of life were higher than those of participants who completed the entire study; this could have resulted in an overestimation of the impact of COVID-19 on the quality of life. Finally, the survey in this study was conducted using a self-reported online questionnaire, which has limitations similar to mailed questionnaires, as we could not confirm whether the participants filled out the questionnaire themselves. Therefore, in small-sample cases, it could impact the accuracy of the results, and our results should be interpreted with caution.

### 4.2. In Future Research

Since our study was conducted after the individuals were discharged and had a small population, caused by the survey being conducted at the end stage of the first COVID-19 outbreak, the change in the quality of life and mental health from hospitalization to discharge could not be observed. Therefore, future research should be conducted earlier to recruit more participants and should commence at the beginning of the patients’ hospitalization period. Additionally, the follow-up period could be extended to observe the long-term changes in individuals after their discharge from the hospital or other quarantine facilities.

## 5. Conclusions

This study investigated the changes in the quality of life and mental health of individuals discharged from the hospital after recovering from COVID-19 within the first week and one month after their discharge. Our findings revealed that one month after their discharge, a high percentage of participants still experienced anxiety problems. Additionally, participants with acute COVID-19 symptoms, abnormal respiratory function, severe conditions during hospitalization, or chronic diseases had lower quality of life and experienced more severe anxiety and stress problems. With the impact of regulations during the pandemic, related issues may persist and be exacerbated in the long term. Therefore, it is crucial to provide COVID-19 symptom management and psychological support to improve patients’ quality of life and reduce their emotional problems. Guidelines or caring telephone calls are suggested methods that benefit discharged patients.

However, since our study included a small sample and was only conducted after discharge, in future studies, we should recruit more participants and start the study with patients who are still in hospital or isolation. Additionally, extending the follow-up period would be beneficial.

## Figures and Tables

**Figure 1 healthcare-12-00488-f001:**
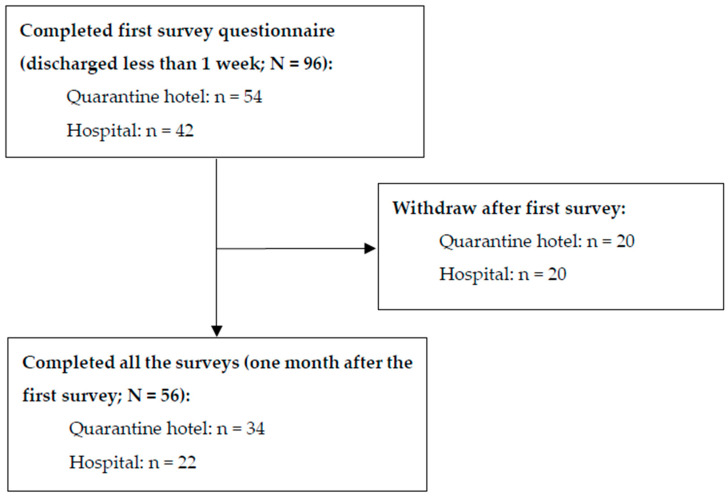
Flow chart of this study.

**Table 1 healthcare-12-00488-t001:** The difference in demographic variables between participants who completed all surveys and those who only participated in the first survey (N = 96).

Variables	Completed All the Surveys	Only Participated in the First Survey	t/X^2^	*p*
	N (%)/Mean ± S.D.	N (%)/Mean ± S.D.		
Age	43.23 ± 13.03	43.22 ± 15.92	−0.00	0.994
Last quarantine facility before discharge			0.69	0.403
Hospital	22 (39.28%)	20 (50%)		
Quarantine hotel	34 (60.71%)	20 (50%)		
Average days in quarantine	17.80 ± 8.30	17.30 ± 9.80	−0.27	0.786
Sex			0.11	0.729
Female	29 (51.78%)	23 (57.50%)		
Male	27 (48.22%)	17 (42.50%)		
Number of chronic diseases			2.65	0.264
0	39 (69.64%)	23 (57.50%)		
1	9 (16.07%)	12 (30.00%)		
2+	8 (14.28%)	5 (12.50%)		
Number of current presenting COVID-19 symptoms			1.51	0.469
0	21 (37.50%)	20 (50.00%)		
1	20 (35.71%)	11 (27.50%)		
2+	15 (26.78%)	9 (22.50%)		
Hospitalized during quarantine			0.30	0.580
No	31 (55.36%)	19 (47.50%)		
Yes	25 (44.64%)	21 (52.50%)		
Admitted to ICU during hospitalization			0.50	0.476
No	49 (87.50%)	32 (80.00%)		
Yes	7 (12.50%)	8 (20.00%)		
Received tracheal intubation during hospitalization			2.50	0.113
No	53 (94.64%)	33 (82.50%)		
Yes	3 (5.36%)	7 (17.50%)		
CAT			0.87	0.350
Normal	42 (75.00%)	34 (85.00%)		
Abnormal	14 (25.00%)	6 (15.00%)		
Quality of life				
Overall quality of life	3.53 ± 0.68	3.77 ± 0.73	1.63	0.105
General health perceptions	3.35 ± 0.84	3.70 ± 0.82	1.98	0.049 *
Physical health	70.79 ± 16.44	72.14 ± 14.69	0.41	0.679
Psychological health	64.36 ± 14.45	67.81 ± 14.97	1.13	0.258
Social relationships	65.17 ± 14.48	69.79 ± 10.95	1.69	0.093
Environment	65.68 ± 14.35	68.28 ± 12.59	0.92	0.359
DASS-21				
Depression	4.25 ± 6.65	3.30 ± 4.16	−0.79	0.426
Anxiety	6.07 ± 6.88	4.75 ± 4.45	−1.06	0.289
Stress	8.03 ± 6.99	5.20 ± 5.23	−2.16	0.032 *

Note—S.D.: Standard Deviation; CAT: Chronic Obstructive Pulmonary Disease (COPD) Assessment Test; DASS-21: Depression Anxiety Stress Scales-21. * *p* < 0.05.

**Table 2 healthcare-12-00488-t002:** Analysis results of the changes between the first and second surveys (N = 56).

Variables	First Survey	Second Survey	t/F/X^2^	*p*
	Mean ± S.D./N (%)	Mean ± S.D./N (%)		
Number of current presenting COVID-19 symptoms			8.07 ^a^	0.017 *
0	21 (37.50%)	36 (64.28%)		
1	20 (35.71%)	12 (21.42%)		
2+	15 (26.78%)	8 (14.28%)		
CAT			0.87 ^a^	0.349
Normal	42 (75.00%)	47 (83.92%)		
Abnormal	14 (25.00%)	9 (16.07%)		
Quality of life				
Overall quality of life	3.54 ± 0.69	3.80 ± 0.64	−3.10 ^b^	0.003 *
General health perceptions	3.36 ± 0.84	3.48 ± 0.79	−1.35 ^b^	0.180
Physical health	70.79 ± 16.45	72.58 ± 16.25	−1.29 ^b^	0.200
Psychological health	64.36 ± 14.45	65.55 ± 17.11	−0.66 ^b^	0.508
Social relationships	65.18 ± 14.49	65.92 ± 17.64	−0.44 ^b^	0.659
Environment	65.68 ± 14.36	67.08 ± 15.46	−1.03 ^b^	0.306
DASS−21				
Depression score	4.25 ± 6.65	4.00 ± 5.85	0.36 ^b^	0.717
Normal	45 (80.36%)	47 (83.93%)	0.06 ^a^	0.805
Abnormal	11 (19.64%)	9 (16.07%)		
Anxiety	6.07 ± 6.89	4.53 ± 5.65	2.60 ^b^	0.200
Normal	38 (67.86%)	43 (76.79%)	0.7 ^a^	0.398
Abnormal	18 (32.14%)	13 (23.21%)		
Stress	8.03 ± 7.00	7.10 ± 7.34	1.06 ^b^	0.292
Normal	49 (87.50%)	47 (83.93%)	0.07 ^a^	0.787
Abnormal	7 (12.50%)	9 (16.07%)		

Note—regarding the DASS-21 results, we conducted an independent *t*-test to analyze the difference in mean scores, and a chi-square test to analyze the difference in the percentages of normal and abnormal scores for each domain. Abnormal emotional problems were determined based on the severity standards for each domain: depression (>9), anxiety (>7), and stress (>14) [24]. ^a^ Analysis using the chi-square test; ^b^ Analysis using the paired *t*-test; S.D.: Standard Deviation; CAT: Chronic Obstructive Pulmonary Disease (COPD) Assessment Test; DASS-21: Depression Anxiety Stress Scales-21. * *p* < 0.05.

**Table 3 healthcare-12-00488-t003:** GEE analysis results of the relationship between quality of life and participants’ demographic variables (N = 56).

Variables	Overall Quality of Life		General Health Perceptions		Physical Health	
	β ± S.E.	*p*	β ± S.E.	*p*	β ± S.E.	*p*
(intercept)	3.48 ± 0.22	<0.001 **	3.47 ± 0.22	<0.001 **	80.87 ± 3.45	<0.001 **
Survey waves						
1	(Reference)		(Reference)		(Reference)	
2	0.21 ± 0.11	0.074	−0.06 ± 0.10	0.562	−1.84 ± 2.18	0.400
Age	0.01 ± 0.01	0.232	0.01 ± 0.004	0.01 *	0.03 ± 0.08	0.660
Sex						
Female	(Reference)		(Reference)		(Reference)	
Male	−0.04 ± 0.11	0.723	−0.003 ± 0.110	0.974	0.57 ± 2.18	0.790
Number of chronic diseases ^a^						
0	-		(Reference)		(Reference)	
1	-		0.08 ± 0.14	0.574	−3.82 ± 2.59	0.140
2+	-		−0.51 ± 0.20	0.011 *	−10.97 ± 4.73	0.020 *
Numbers of current presenting COVID-19 symptoms ^a^						
0	-		(Reference)		(Reference)	
1	-		−0.34 ± 0.13	0.008 **	−3.71 ± 2.52	0.140
2+	-		−0.63 ± 0.14	<0.001 **	−13.77 ± 3.27	<0.001 **
Received treatment in ICU during hospitalization ^b^						
No	-		(Reference)			
Yes	-		−0.21 ± 0.24	0.380	-	-
Used oxygen supply cartridges during hospitalization ^c^						
No	(Reference)		(Reference)		(Reference)	
Yes	−0.15 ± 0.15	0.325	−0.39 ± 0.23	0.098	0.71 ± 3.11	0.820
Underwent tracheal intubation during hospitalization ^d^						
No	-		-		(Reference)	
Yes	-		-		−21.26 ± 4.76	<0.001 **
CAT ^e^						
Normal	(reference)		(reference)		(reference)	
Abnormal	−0.64 ± 0.11	<0.001 **	−0.66 ± 0.15	<0.001 **	−15.48 ± 3.48	<0.001 **
**Variables**	**Psychological**		**Social relationships**		**Environment**	
	**β ± S.E.**	* **p** *	**β ± S.E.**	* **p** *	**β ± S.E.**	* **p** *
(intercept)	63.97 ± 4.39	<0.001 **	58.92 ± 5.25	<0.001 **	63.30 ± 4.60	<0.001 **
Survey waves						
1	(Reference)		(Reference)		(Reference)	
2	−0.33 ± 2.56	0.895	−0.24 ± 2.85	0.931	0.91 ± 2.65	0.732
Age	0.14 ± 0.09	0.123	0.16 ± 0.11	0.162	0.10 ± 0.09	0.297
Sex						
Female	(Reference)		(Reference)		(Reference)	
Male	−1.62 ± 2.59	0.532	1.74 ± 2.90	0.546	2.31 ± 2.66	0.383
Underwent tracheal intubation during hospitalization ^d^						
No	(Reference)		-		-	
Yes	−17.56 ± 4.75	<0.001 **	-		-	
CAT ^e^						
Normal	(Reference)		(Reference)		(Reference)	
Abnormal	−17.07 ± 2.75	<0.001 **	−10.22 ± 3.09	<0.001 **	−11.45 ± 3.55	0.001 **

Note—the variables included in each model for adjustment are survey wave, age, sex, and those variables that significantly affected the domains of the quality of life. ^a^ This analysis was conducted using a one-way ANOVA, which revealed a significant effect of this variable on the ‘General health perceptions’ and ‘Physical health’ domains of quality of life. ^b^ This analysis was conducted using independent *t*-tests, which revealed a significant effect of this variable on the ‘General health perceptions’ domain of quality of life. ^c^ This analysis was conducted using independent *t*-tests, which revealed a significant effect of this variable on the “Overall Quality of Life”, “General Health Perceptions” and “Physical Health” domains of the quality of life. ^d^ This analysis was conducted using independent *t*-tests, which revealed a significant effect of this variable on the “Physical health” and “Psychological” domains of quality of life. ^e^ This analysis was conducted using independent t-tests, which revealed a significant effect of this variable on all three domains of the quality of life. S.D.: Standard Deviation; ICU: Intensive Care Unit; CAT: Chronic Obstructive Pulmonary Disease (COPD) Assessment Test. * *p* < 0.05. ** *p* < 0.001.

**Table 4 healthcare-12-00488-t004:** GEE analysis results of the relationship between emotional problems and participants’ demographic variables (N = 56).

Variables	Anxiety		Stress	
	β ± S.E.	*p*	β ± S.E.	*p*
(intercept)	1.44 ± 1.56	0.355	4.72 ± 1.98	0.017 *
Survey waves				
1	(Reference)			
2	−0.39 ± 1.00	0.693	−0.16 ± 1.19	0.887
Age	0.03 ± 0.03	0.305	0.04 ± 0.04	0.366
Sex				
Female	(Reference)			
Male	0.21 ± 1.053	0.840	−0.76 ± 1.23	0.532
Numbers of current presenting COVID−19 symptoms ^a^				
0				
1	1.10 ± 1.53	0.472	−0.97 ± 1.74	0.577
2+	3.65 ± 1.34	0.006 **	2.92 ± 1.50	0.051
CAT ^a^				
Normal	(Reference)			
Abnormal	5.88 ± 1.78	<0.001 **	5.98 ± 2.11	0.004 *

Note—the variables included in each model for adjustment are survey wave, age, sex, and those variables that significantly affected the domains of the quality of life. Depression was not included in this table because no variables were found to significantly affect depression. ^a^ This analysis was conducted using independent *t*-tests, which revealed a significant effect of this variable on the “Anxiety” and “Stress” domains of mental health. S.D.: Standard Deviation; CAT: Chronic Obstructive Pulmonary Disease (COPD) Assessment Test. * *p* < 0.05. ** *p* < 0.001.

## Data Availability

The data that support the findings of this study are available from the corresponding author upon reasonable request.

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
