# Peer review of "Quality of Life and Emotional Problems of COVID-19 Patients after Discharge: A One-Month Longitudinal Study"

_healthcare, 2024, doi:10.3390/healthcare12040488_

Round 1

Reviewer 1 Report

Comments and Suggestions for Authors

The article is written quite well and accurately presents the design, method and results. The introduction is sound and well presented. In the method section It would be interesting to know more about the addition on the QoL questionnaire regardind Taiwan cultural habits. Overall the tables MUST be improved. Please adjust font, alignment and readibility of inserted data, as it is quite scrambled and a source of confusion; while the results are explained clearly in the appropriate section, tables create some problems.

In the conclusion It would be appropriate to add more depth to the prblem of support and aid to patients that experienced quarantine, even citing more articles.

Reviewer 2 Report

Comments and Suggestions for Authors

This manuscript has merits, but there are several places requiring clarification and improvement. Below are my comments:

11. Cite references for Taiwan’s COVID-19 outbreak data. For example, “In Taiwan, the COVID-19 local outbreak began in May 2021 and continued until June 30, 2021, wherein 14748 individuals were infected”.

22. Author mentioned the instrument of WHOQoL-BREF was translated into Taiwanese? How about the other instruments used in this study? The participants were administered using English version? Also, is translated version of WHOQoL-BREF (Yao et al.) translated into Taiwanese or Traditional Chinese?

33.  The author reported the range of Cronbach’s alpha (0.85-0.98) for respiratory function. Is that the range between subscales? No subscales were mentioned in Respiratory function section.

44. Were the Cronbach alphas reported in “Materials and Methods” section from the original study or this current study? 

55. It is unclear where are the results of the 3rd analyses mentioned in Data analysis. Indicate what are the confounding variables included in the GEE analyses? And why GEE analyses were used? 

66.  Include sample size in each table.

77. Table 1, in table heading, the notation of chis-square test missed the power of 2.

88. Table 2 and Table 4, ** p<0.001 was never used in table, but it was included in table footnote.

99. Table 3, the S.E. of age for general health perceptions is not shown. “0.00” should be changed.

110. Table 4, reference groups should be indicated.

111. In the discussion section, the authors did not mention this is a study conducted in Taiwan. In addition, it is unclear what is the normal population in the sentence “….in terms of quality of life, when compared to the report in the Taiwanese version….., the score was not lower than that of the normal population.” The normal population in Taiwan? The normal population in other areas?

112. Line 305, indicate which year “after August”?

113. There are other limitations that should be included such as sample size and the generalizability. The author mentioned data was from one teaching hospital, but it should be indicated where this hospital is located.  

Reviewer 3 Report

Comments and Suggestions for Authors

Dear authors,

It is a pleasure reviewing your work. To improve it, please follow the instructions below:

Title:

  • Original Title: "Longitudinal study: A follow-up study on the quality of life and emotional problems of COVID-19 patients within one week and after one month of discharge."
  • Suggested Improvements:
    • Conciseness: The Title is somewhat lengthy and could be more concise to enhance impact and readability. Consider shortening it while maintaining the essence of the study. For example: "Quality of Life and Emotional Well-being Post-COVID-19: A One-Month Longitudinal Study".
    • Clarity: Ensure the Title reflects the core focus and methodology of the paper. If the longitudinal aspect is crucial, emphasize it without making the Title too wordy.
    • Specificity: If possible, specify the "quality of life" or "emotional problems" the study mainly focuses on.

Abstract:

  • Structure & Content:
    • Aim and Objective: The aim and objective are clearly stated, but they could be merged for brevity and to avoid redundancy. The time frame and patient isolation details are proper, but consider if they can be integrated more succinctly.
    • Methods: The description of the methodology is clear, explaining the longitudinal design and the use of online questionnaires. It might be beneficial to briefly mention the main variables or scales used in the questionnaires for quality of life and emotional problems.
    • Results: Results are specific and informative, providing a good snapshot of the study's findings. However, highlight the most impactful or surprising result to draw the reader's attention.
    • Conclusion: The conclusion ties back to the aim nicely. It might be enhanced by briefly suggesting potential practical applications of the findings or implications for future research or healthcare policies.
  • Language & Style:
    • Clarity and Flow: Ensure the abstract is understandable to specialists and non-specialist readers. Some sentences are pretty long and could be broken down for clarity.
    • Tone: Maintain a scholarly tone throughout, ensuring that it reflects the serious and analytical nature of the study.
  • Keywords: "COVID-19; QoL; mental health" are relevant. Consider if there are any other key terms specific to your study's focus (e.g., "longitudinal," "post-discharge," or specific emotional problems like "anxiety") that could aid in searchability.

Introduction

Strengths:

  • Provides a transparent background on COVID-19, its impact, and the situation in Taiwan.
  • Effectively establishes the importance of studying health-related quality of life and mental health in COVID-19.

Areas for Improvement:

  1. Contextual Depth: While the introduction outlines the COVID-19 situation in Taiwan, it could benefit from briefly discussing global context or comparisons to illustrate the broader relevance of the study.
  2. Flow and Clarity: Some sentences are long and could be broken into shorter, more straightforward statements to improve readability.
  3. Literature Review: Expand on the existing studies to provide a more comprehensive overview of the current knowledge and gaps your study aims to fill.

Methods and Materials

Strengths:

  • Detailed description of the participant selection, data collection process, and research tools.
  • Using validated instruments (CAT, WHOQoL-BREF, DASS-21) adds credibility.

Areas for Improvement:

  1. Clarity in Selection Criteria: Clarify why the specific age limit and exclusion criteria were chosen. If there are particular reasons, they should be briefly mentioned.
  2. Incentives: Discuss the ethical considerations of providing incentives for survey completion. Ensure it is clear that the incentive does not bias the responses or selection of participants.
  3. Description of Tools: While the tools are described well, briefly discussing why these particular tools were chosen over others could provide more insight into the study design.

Data Analysis

Strengths:

  • A comprehensive analytical approach using a range of statistical tests and methods.
  • Explicit mention of the software used for analysis.

Areas for Improvement:

  1. Technical Terms: Some readers might not be familiar with all the statistical methods mentioned. A brief, non-technical explanation of why specific tests are used could be helpful.
  2. Predictive Modeling: If the study aims to predict or identify risk factors, discussing any predictive modeling or multivariate analysis would be beneficial.

Ethical Considerations

Strengths:

  • Thoroughly addresses consent, permissions for instruments, and ethical approval.

Areas for Improvement:

  1. Participant Anonymity and Data Protection: Elaborate on how participants' data is protected and anonymized, especially since the study involves potentially sensitive health information.
  2. Potential Biases: Discuss any potential biases or ethical considerations related to the online nature of the questionnaire and how they were mitigated.

General Suggestions:

  • References: Ensure all claims, especially statistical findings and health-related statements, are backed by up-to-date and reputable sources.
  • Consistency: Check for consistent formatting, especially in the references and citations.
  • Language and Grammar: Ensure the document is free from grammatical errors and that the language is professional and appropriate for an academic paper.

Results Section

Strengths:

  • Detailed presentation of statistical analysis and findings.
  • A clear distinction between groups and time points.
  • Use of tables to organize and present data.

Areas for Improvement:

  1. Clarity and Readability:
    • Break down complex sentences and ensure each finding is presented clearly and succinctly.
    • Avoid jargon where possible or provide brief explanations for technical terms.
  2. Presentation of Data:
    • Ensure tables are well-organized and easy to understand. Consider adding labels or notes to clarify what specific terms and abbreviations mean.
    • Include visual aids like graphs or charts where appropriate to enhance understanding of the trends and comparisons.
  3. Drop-out Analysis:
    • Address the drop-out rate more thoroughly. Out of 98 initial participants, only 56 completed the final questionnaire. Discuss potential reasons for this drop-out rate and how it might affect the results.
  4. Statistical Significance and Practical Significance:
    • While statistical significance is noted, discuss the practical relevance of the findings. For example, what does a change in the quality of life or stress score mean in real terms for a patient?
    • Discuss any results close to being statistically significant and what that might imply.
  5. Contextualize Findings:
    • Compare and contrast your findings with existing literature. Are your results consistent with other studies, or do they offer new insights?
    • If there are discrepancies or unexpected findings, discuss potential reasons and implications.
  6. Discuss Limitations and Implications:
    • Every study has limitations. Discuss your research, such as sample size, selection bias, or measurement tools.
    • Explain how these limitations might affect the interpretation of the results and what future research might address them.
  7. Enhance the Discussion of Demographics:
    • Provide more context or discussion around demographic findings. For instance, if there were no significant differences in demographics between groups, what might this imply about the generalizability of the results?
  8. Address Non-Significant Results:
    • Discuss why some expected significant changes (e.g., quality of life domains) were not observed. This could include speculation on the nature of recovery, the measures used, or other external factors.
  9. Subgroup Analysis:
    • If the data allows, consider conducting subgroup analyses to see if certain groups (e.g., by age, sex, severity of symptoms) have different outcomes or trends.
  10. Enhanced Conclusion within Results:
    • Briefly summarize the most critical findings at the end of the results section to provide a clear takeaway for the reader before moving on to the discussion.

Discussion Section

Strengths:

  • The discussion comprehensively compares the study's results and existing literature.
  • It acknowledges various factors that could influence the psychological and quality of life outcomes of COVID-19 patients.
  • It rightly notes the implications of the findings for future interventions and policy.

Areas for Improvement:

  1. Contextualization and Comparison:
    • While the study compares its findings to existing literature, it could benefit from a deeper analysis of why specific results align or differ from other studies. Discuss the possible reasons behind contradictory findings in more detail.
    • Provide a broader context of how these findings fit into the larger body of COVID-19 research.
  2. Specificity in Recommendations:
    • The suggestion for monitoring and interventions is valuable. However, more specific recommendations based on the findings would enhance the discussion. For example, what type of psychological support or symptom management strategies would be most effective?
  3. Addressing Limitations:
    • The limitations are well-noted, but consider discussing how these limitations might specifically impact the study's conclusions. Also, suggest ways future research could overcome these limitations.
    • Discuss potential biases or errors that might arise from using self-reported online questionnaires and how they might influence the results.
  4. Broader Implications:
    • Discuss the broader implications of the study's findings for public health policies, healthcare practices, or future epidemic responses.
    • How can these findings inform the design of mental health and social support services for post-COVID-19 patients?
  5. Clarity and Readability:
    • Ensure that complex sentences are clear and concise to enhance readability.
    • Avoid overuse of technical jargon or explain terms that might not be widely understood.
  6. Future Research Directions:
    • Suggest areas where further research is needed based on the study's findings and limitations. What unanswered questions remain?
    • Discuss how future studies could build on this work to provide more insights into the long-term effects of COVID-19.
  7. Inclusion of Additional Theories or Models:
    • Incorporate relevant psychological or health theories to explain the observed effects and support the discussion. This could provide a more robust theoretical framework for understanding the results.
  8. Greater Emphasis on Participant Characteristics:
    • Discuss how different characteristics of the participants (e.g., age, sex, pre-existing conditions) might influence the generalizability of the findings and the effectiveness of proposed interventions.
  9. Clarify the Nature of Contradictions:
    • When mentioning contradictory results, clarify whether these are due to differences in study design, population, disease severity, or other factors.
  10. Address the Unexpected or Unexplained:
    • If there were any unexpected or unexplained findings, discuss these in more detail. Speculate on potential reasons and how they might be explored in future research.

Conclusion Section

Strengths:

  • The conclusion briefly summarizes the study's main findings regarding the impact of COVID-19 on the quality of life and mental health of patients post-discharge.
  • It acknowledges the specific factors that were found to be associated with worse outcomes, such as severe symptoms, respiratory function, and chronic diseases.

Areas for Improvement:

  1. Broader Implications:
    • Discuss the broader implications of your findings for public health, healthcare delivery, or policy. How can these insights be used to support better recovering COVID-19 patients?
    • Mention any potential for these findings to inform future research or health strategies, particularly in managing post-COVID conditions.
  2. Specificity in Recommendations:
    • The recommendation for symptom management and psychological support is valuable. However, more specific suggestions based on your findings would enhance the conclusion. For instance, what particular forms of psychological support or symptom management strategies would be beneficial?
  3. Reflect on Limitations:
    • Briefly mention how the study's limitations might influence the findings' interpretation and the recommendations' generalizability.
  4. Call for Future Research:
    • Encourage further investigation into the areas where your study has found significant results or where questions remain. This can guide future studies to build upon your work.
  5. Clarity and Conciseness:
    • Ensure that the conclusion is clear and concise. Each sentence should add value and insight based on the study's findings.
  6. Long-Term Perspective:
    • If applicable, comment on the potential long-term implications of your findings. How might the quality of life and mental health of COVID-19 survivors be affected in the longer term?
  7. Reflect on Unexpected Findings:
    • If there were any unexpected or particularly notable findings, briefly mention these and their potential implications or the need for further exploration.
  8. Highlight Novelty or Importance:
    • If your study has revealed new insights or contradicted previous assumptions, highlight the importance of these revelations in the conclusion.

References Section

Strengths:

  • The references are diverse, including articles from various reputable journals and sources.
  • URLs and DOIs are provided for most sources, facilitating easy access to the original articles.

Areas for Improvement:

  1. Consistency in Formatting:
    • Ensure consistent formatting across all references. This includes presenting authors' names, journal names, volume and issue numbers, page ranges, and dates. Consistency makes the reference section more professional and easier to navigate.
  2. Accuracy of Information:
    • Double-check each reference for accuracy. Ensure that author names, article titles, journal names, and publication years are correctly stated and formatted.
  3. Accessibility of Links:
    • For URLs provided, ensure they are accessible and lead directly to the cited document. Test each link to confirm it works and leads to the correct document.
  4. Standardization of Journal Abbreviations:
    • If you use abbreviations for journal names, ensure they are standardized and recognizable. Alternatively, use full journal names for clarity.
  5. Updating References:
    • Ensure all references are as up-to-date as possible. If more recent studies or data are available, consider including them to strengthen the paper's relevance and authority.
  6. Inclusion of All Relevant Sources:
    • Verify that the paper's studies, theories, and data are correctly cited in the reference section.
  7. Publisher and Location:
    • Certain types of references, especially books, include the publisher and location of publication.
  8. Avoiding Redundancy:
    • Ensure that each reference is unique and cited appropriately in the text. Duplicate references or unnecessary citations should be removed.
  9. Citation Standards:
    • Adhere to the specific citation style guide (e.g., APA, MLA, Chicago) recommended by your publication or field of study. This includes in-text citations and the reference list.

Yours sincerely

Comments on the Quality of English Language

Dear Authors,

The English writing in the provided text is generally clear and maintains a formal, academic tone appropriate for a scholarly article. However, there are areas where the language could be refined for clarity, coherence, and conciseness. Here are some specific observations and suggestions for improvement:

  1. Title and Subtitles:
    • The title is quite long and could be more concise. It is usually beneficial to make titles straightforward while conveying the essence of the study.
  2. Author Affiliations:
    • The affiliations and contact details are appropriately listed, but ensure that the formatting is consistent and professional across all author entries.
  3. Abstract:
    • The abstract is generally well-written. However, sentences could be more concise. For example, "investigate changes in the quality of life and mental health" could be shortened to "investigate changes in quality of life and mental health" without losing meaning.
  4. Introduction:
    • There are minor grammatical issues, such as the unnecessary comma in "the disease quickly became a pandemic [3].".
    • Ensure consistency in the use of terms. For instance, "COVID-19 virus" is typically referred to as just "COVID-19" or "SARS-CoV-2".
  5. Consistency and Formatting:
    • Ensure the text doesn't break inappropriately across lines, mainly for hyphenated words or numerical data.
  6. Clarity and Precision:
    • Some phrases could be more precise.
    • Avoid passive voice where possible to make statements more direct and vigorous.

These are just some examples to take a starting point for a language review.

Yours sincerely

Reviewer 4 Report

Comments and Suggestions for Authors

This is a purely descriptive observational study following the application of quality of life tests and with a very small sample size that severely limits the generalisability of any conclusions.

On the other hand, in the material and methods it is indicated that payment was made to the patients who completed the questionnaires, which is an ethically unacceptable practice and generates evident response biases.

I do not recommend its publication.

Author Response

Thanks for the suggestions.

Due to time constraints and the size of the hospital, we are limited in the sample size and cannot recruit more participants for this study. Nevertheless, we have addressed confounding factors through analysis methods such as modeling by GEE. Moreover, similar research is rare in Taiwan. Therefore, we are confident that our research holds significant value.

In terms of incentives, we offer a 100 NTD (3.23 USD) gift to encourage full participation in the study. This amount is considered a modest sum in Taiwan and is commonly used in surveys and research. Additionally, according to Stanley et al. (2020), even incentives approaching 5 USD in online surveys show no significant response bias. Therefore, we believe that the incentive we provide would not have an impact on the results.

Round 2

Reviewer 4 Report

Comments and Suggestions for Authors

After reviewing the new version of the paper and the authors' justification regarding the sample size and the "gifts" to patients for participating in the survey, I am not convinced by their arguments and they reinforce my previous position that I do not recommend publication of the paper in a Q1-Q2 journal. Perhaps it could be published in another journal with less impact.

Author Response

My apologies, I mistakenly used the term 'gift' to describe the incentive fee for completing the questionnaire. It should be clarified that it was actually a gift certificate. This incentive fee serves more as compensation for the expenses incurred by participants in filling out the questionnaire and traveling to an area with Internet access. 100 NTD is not a substantial amount in Taiwan. If someone takes a taxi to a location and connects to the internet to complete the questionnaire, the round-trip transportation cost alone is at least 190 NTD. Therefore, this incentive may not be very enticing, and it could also lead to a lower response rate in the second survey of this study.